# The Structural Origin of Attention Sink:
# Variance Discrepancy, Super Neurons, and Dimension Disparity

**Siquan Li**[1]   **Kaiqi Jiang**[2]   **Jiacheng Sun**[2]   **Tianyang Hu**[1]

## Abstract

Despite the prevalence of the attention sink phenomenon in Large Language Models (LLMs), where initial tokens disproportionately monopolize attention scores, its structural origins remain elusive. This work provides a *mechanistic explanation* for this phenomenon. First, we trace its root to the value aggregation process inherent in self-attention, which induces a systematic variance discrepancy. We further demonstrate that this discrepancy is drastically amplified by the activation of super neurons within Feed-Forward Network (FFN) layers. Specifically, the channel-sparse down-projections trigger a dimension disparity of the first-token representation, necessitating the formation of attention sinks as a structural anchor. Then, we validate this causal chain through two controlled interventions: (i) isolating the aggregation effect via attention mask modifications and (ii) amplifying the variance of targeted token representations. Both interventions can replicate attention sinks at arbitrary positions. Our mechanistic understanding offers a foundation for the systematic control of sink formation. Finally, as a proof of concept, we propose *head-wise RMSNorm*, an architectural modification that stabilizes value aggregation outputs during pre-training. Our experiments demonstrate that restoring statistical parity across positions significantly accelerates convergence. The code is available at https://github.com/Siquan-Li/Head-wise-RMSNorm.

## 1. Introduction

Attention sinks are a recurring feature of decoder-only transformers: across layers and inputs, a small set of tokens, most notably the initial token, can receive disproportionately large attention despite limited semantic relevance [25, 7, 3]. From a functional standpoint, this phenomenon is a double-edged sword: while it enables efficient KV cache compression strategies like streaming generation [10, 26] and mitigates over-smoothing [2], it also gives rise to pathological behaviors that reflect deeper architectural issues, such as activation outliers [14, 21], representation collapse [13, 22], and optimization abnormalities [6]. Existing literature offers diverse hypotheses for its formation, including but not limited to the Softmax operator's need for a "sink" for residual probability mass [27], positional mechanisms [29], or spectral subspaces [4]. However, the fundamental causal chain that dictates why the initial token is consistently selected as the structural anchor remains to be elucidated.

In this work, we bridge this gap by tracing the structural origin of attention sinks to the *variance discrepancy* (see detailed definition in Section 3.1) inherent to the *value aggregation* process of self-attention. Specifically, under causal masking, the initial token attends only to itself, while subsequent tokens aggregate information from an expanding context. Consequently, the initial token—exempt from this averaging—persists as a high-variance outlier. Through comprehensive investigation into the internal propagation of transformers, we find that these outliers are preserved by the output projection in the attention module that later activate *super neurons* within the FFN, triggering massive activations and a subsequent *dimension disparity* of token representations. Propagated through residual connections and normalization layers, these distortions ultimately dominate the query-key dot products, necessitating the formation of the attention sink. A schematic overview of this propagation chain is illustrated in Figure 1.

To validate the causal link from variance discrepancy to attention sink, we conduct two controlled interventions to mimic the variance property of the initial token: (i) modifying the attention mask to isolate the aggregation effect, and (ii) amplifying the variance of arbitrary token representations. Both interventions can induce attention sink at

[1]The Chinese University of Hong Kong, Shenzhen [2]Huawei Foundation Model Department. Correspondence to: Tianyang Hu <hutianyang@cuhk.edu.cn>.

*Proceedings of the 43rd International Conference on Machine Learning*, Seoul, South Korea. PMLR 306, 2026. Copyright 2026 by the author(s).

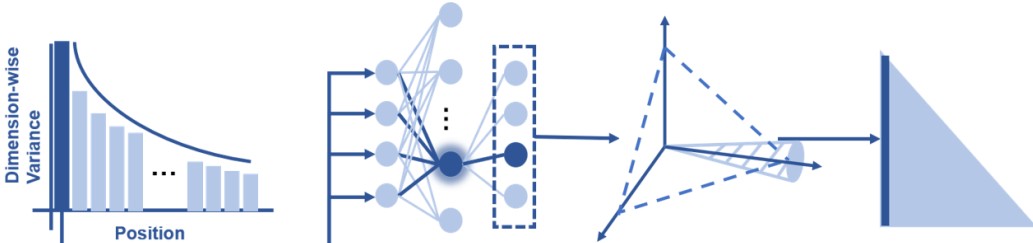

**Figure 1. Schematic Overview of the Attention Sink Mechanism.** Value aggregation causes dimension-wise variance decay for subsequent tokens, while the first token acts as a high-variance outlier. This discrepancy is preserved by output projections, activating super neurons in FFNs. Subsequently, the channel-sparse down-projections induce dimension disparity, resulting in the attention sink.

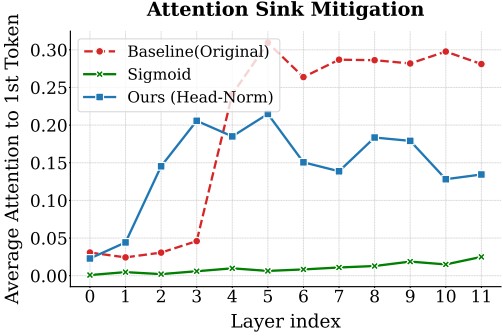

**Figure 2. Mitigation of Attention Sinks.** Comparison of the averaged attention to the first token across layers. The Baseline (red) exhibits attention sink from the 5th layer, whereas both Sigmoid attention (green) and our Head-wise RMSNorm (blue) method successfully suppress this artifact.

arbitrary positions, establishing the causal effect of variance discrepancy on attention sink formation.

To verify our mechanistic insight, we first test an existing variant: replacing Softmax with a *Sigmoid activation*. Without the sum-to-one constraint, the first token is no longer a high-variance outlier. As expected, attention sinks are significantly mitigated in the pretraining process (see Figure 2). Building on this, we further introduce *head-wise RMSNorm*, a novel modification designed to stabilize value aggregation outputs. Pretraining experiments demonstrate that this targeted intervention not only suppresses attention sinks but also accelerates pretraining convergence.

Our results suggest that the attention sink is not an inevitable byproduct of scaling, but a controllable architectural property, providing a new foundation for the design of more stable and interpretable transformers.

## 2. Preliminary

In this work, we focus on the transformer decoder architecture, which serves as the backbone for modern LLMs. A standard decoder block consists of two primary sub-

layers: a self-attention and an FFN, both employing pre-normalization and residual connections.

Formally, let $\mathbf{x}_l \in \mathbb{R}^{T \times d}$ denote the input hidden states for the $l$-th layer, where $T$ is the sequence length and $d$ is the hidden dimension. The forward propagation within a single layer is defined as:

$$\mathbf{h}_l = \mathbf{x}_l + \text{Attention}(\text{Norm}(\mathbf{x}_l)),$$
$$\mathbf{x}_{l+1} = \mathbf{h}_l + \text{FFN}(\text{Norm}(\mathbf{h}_l)).$$

***Self-Attention*** The self-attention mechanism [24] aggregates context by computing a convex combination of value vectors. Let $A_{i,j}$ denote the normalized attention score between token $i$ and $j$, and $\mathbf{V}_{j,k}$ represent the value state at position $j$ and dimension $k$. The *value aggregation* for the $i$-th token is expressed as:

$$o_{i,k} = \sum_{j=0}^{i} A_{i,j} \cdot \mathbf{V}_{j,k}, \quad \text{subject to} \quad \sum_{j=0}^{i} A_{i,j} = 1. \quad (1)$$

The aggregated states are subsequently projected by $\mathbf{W}_O \in \mathbb{R}^{d \times d}$ to form the layer output.

***Feed-Forward Network*** Llama-2 [23] employs the SwiGLU variant [20] for its FFN. It consists of three linear projections and a SiLU activation function. Given an input $\mathbf{x} \in \mathbb{R}^d$, the output is computed as:

$$\text{FFN}(\mathbf{x}) = (\text{SiLU}(\mathbf{x}\mathbf{W}_{gate}) \odot \mathbf{x}\mathbf{W}_{up})\mathbf{W}_{down}, \quad (2)$$

where $\odot$ represents element-wise multiplication. The weights are parameterized as $\mathbf{W}_{gate}, \mathbf{W}_{up} \in \mathbb{R}^{d \times d_f}$ and $\mathbf{W}_{down} \in \mathbb{R}^{d_f \times d}$, where $d_f$ is the intermediate dimension.

## 3. Structural Origins of Attention Sinks

Prior research has established that attention sinks are *semantically irrelevant*, persisting even in sequences of random tokens [9]. In this work, we investigate the internal evolution

of token representations and structural origins of attention sinks. Our primary analysis centers on *Llama-2-7B* [23]. To ensure generality, we also validate our findings on other open-source LLMs in Appendix A.

To understand the nature of attention sinks, we investigate *where* the attention sink emerges within the LLMs and what might be a *trigger* before it happens.

***Invariant layer-wise onset*** We track the attention patterns across all layers on WikiText-2. As shown in the blue curve of Figure 3, we observe a consistent pattern: the attention weight on the initial token remains low in the initial layers but exhibits a sudden spike around layer 2. This structural consistency suggests that the onset of the sink is a fixed property of the model's depth and architecture, occurring as soon as internal statistics accumulate to a critical point.

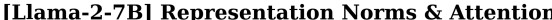
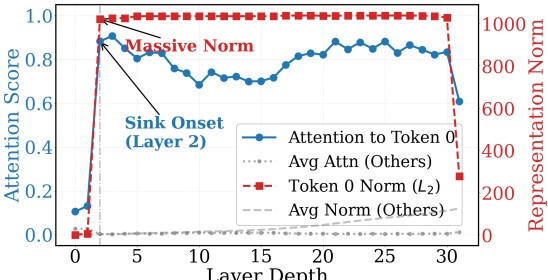

*Figure 3.* **Layer-wise evolution of attention sink and representation norms.** We plot the attention score of the first token (left axis, blue) and its input representation $l_2$-norm (right axis, red) for Llama-2. The synchronized spike indicates that the arrival of a high-norm representation triggers the attention sink.

***The precursor: massive representation norms*** To identify *what* triggers this sudden spike, we analyze the hidden states entering each layer. We track the $l_2$-norm of the representation for the first token. As shown in the red curve of Figure 3, we observe a distinct anomaly: the $l_2$-norm of the representation for the first token increases sharply at the exact same layer where the attention sink emerges. This synchronization implies a direct link between the two phenomena, suggesting that the massive representation norm is a critical precursor to the attention sink.

### 3.1. Value Aggregation Introduces Positional Variance Discrepancy

Attention sink is position-specific. Given that FFNs, Layer-Norm [1], and residual connections [11] operate identically across positions, the root cause of the first-token anomaly implies a mechanism unique to the *self-attention* process.

We focus on the attention mechanism, specifically its *value aggregation* process, as formulated in Eq. (1). The causal mask induces a structural difference: the initial token ($i = 0$)

strictly attends to itself ($a_{0,0} = 1$), while subsequent tokens ($i > 0$) aggregate previous value vectors. As a result, the variance of the value vectors tends to decay as $i$ increases, rendering the first token as an outlier [13].

We validate this positional variance discrepancy in *Llama-2-7B*. Since the attention sink consistently emerges at layer 2, we investigate the dimension-wise variance of hidden states in Layer 1 immediately after value aggregation. Specifically, we compute the variance for each hidden dimension across the batch and report the average for each position. Crucially, to eliminate the bias of a fixed beginning-of-sentence (BOS) token—which would mathematically yield zero variance—we utilize sequences of fully random tokens.

As shown in Figure 4, there is a clear *variance decay* pattern: the average dimension-wise variance exhibits a sharp drop as the token position index increases. The first token retains a significantly higher variance across its dimensions compared to the rest of the sequence.

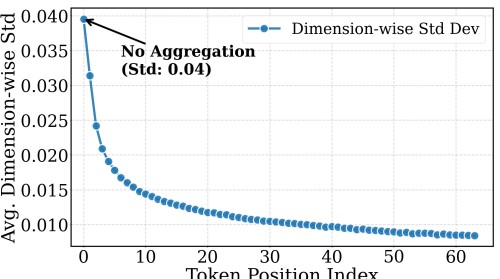

*Figure 4.* **Mean dimension-wise variance discrepancy.** We plot the mean dimension-wise standard deviation of hidden states immediately after (Layer 1) value aggregation. Position 0 exhibits exceptionally high variance and subsequent tokens show a sharp variance decay.

### 3.2. Causal Effect of Variance Discrepancy on Attention Sink

Given that the first token is an outlier in terms of variance, we investigate its causal link to the observed attention sink. To this end, we devise two interventions aiming to introduce attention sinks at *arbitrary* locations by manipulating the variance discrepancy.

***Attention Mask Intervention*** Consider modifying the attention mask of the $k$-th token to block it from attending to any preceding positions ($j < k$). This forces the token to attend only to itself, effectively simulating the state of the initial token. Attention mechanism of subsequent tokens ($t > k$) is left unchanged. Through this intervention, we can mimic the variance behavior of the first token at any position and evaluate the attention pattern. Experimental results on *Llama-2-7B* are shown in Figure 5. When aggregation is blocked at index $k = 10$ (red line), it immediately

becomes a new attention sink. This supports that the sink phenomenon is not tied to the absolute position 0, but a consequence of *unaggregated high variance representations*.

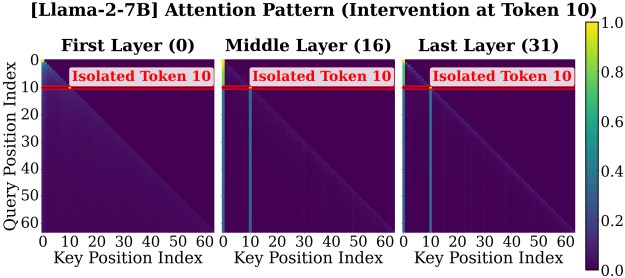

*Figure 5.* **Causal validation via mask intervention.** We visualize the average attention score received by each token on Llama-2. The *red line* shows the result of blocking aggregation at *index 10*, which causes it to transform into a new attention sink.

***Direct Variance Amplification*** We can also mimic the initial-token variance behavior at any position $k$ by directly amplifying the corresponding variance. First, we compute the global mean of the value vectors, denoted as $\boldsymbol{\mu}^{(l)}$, at each layer $l$ using random tokens. This is averaged over both batch size and sequence length.

Then, for an arbitrary token at index $k$, we amplified its aggregated output's variance using a scalar $\lambda > 0$:

$$\mathbf{o}'^{(l)}_k = \boldsymbol{\mu}^{(l)} + \lambda \cdot (\mathbf{o}^{(l)}_k - \boldsymbol{\mu}^{(l)}).$$

As shown in Figure 6, increasing $\lambda$ under our mean-centered amplification consistently transforms the $k$-th token into a dominant attention sink.

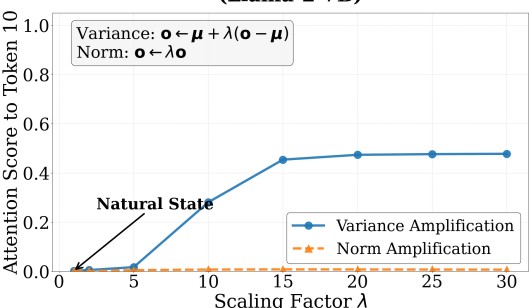

*Figure 6.* **Inducing attention sinks via variance amplification.** We apply a factor $\lambda$ to amplify the variance of an arbitrary token (index 10). Increasing $\lambda$ directly increases the attention score received by the token. A control experiment shows that merely scaling the representation norm fails to induce such a sink.

To rule out the possibility that the induced sink is merely an artifact of an enlarged representation norm (which could trivially inflate the query-key dot product), we conduct a control experiment by directly scaling the output vector

by the same factor: $\lambda \cdot \mathbf{o}^{(l)}_k$. Figure 6 shows that simply scaling the representation norm fails to reproduce the sink formation, demonstrating that the *magnitude of variance* is the critical factor triggering the attention sink.

## 4. Mechanism Analysis: From Variance Discrepancy to Attention Sinks

Having established that high variance caused by the lack of aggregation drives the attention sink, we now examine how this anomaly propagates through the transformer. We focus our analysis on *Llama-2-7B* to trace the internal chain reaction. Specifically, we identify a multi-stage propagation chain that transforms the initial variance discrepancy into the final attention sinks:

1. **Output projection preserves variance discrepancy (Sec. 4.1):** The output projection ($\mathbf{W}_O$) structurally preserves the variance discrepancy, injecting it directly into the residual stream.

2. **Selectively activated by FFN super neurons (Sec. 4.2):** Following the pre-FFN RMSNorm, the first token's distinct representation selectively activates *super neurons* in the FFN, causing massive activations that are then channeled through the sparse down-projection, resulting in extreme dimension disparities in the output.

3. **Locking via QK projection (Sec. 4.3):** The massive dimension disparity persists through the subsequent pre-attention RMSNorm, effectively "locking" the Query-Key projection in the next layer, which compels the attention mechanism to sink to the first token.

In the following subsections, we empirically verify each stage of this propagation chain.

### 4.1. Output Projection Preserves Variance Discrepancy

Since the representations carrying dimension-wise variance discrepancy are first processed by the output projection, a critical question arises: does the output projection $\mathbf{W}_O$ suppress or preserve the variance discrepancy generated by the lack of aggregation? To answer this, we investigate the structural alignment between $\mathbf{W}_O$ and the high-variance dimensions of the first token.

***Structural Alignment Analysis*** We hypothesize that $\mathbf{W}_O$ is biased to amplify dimensions where the first token has high variance. Let $\boldsymbol{\sigma}_{in} \in \mathbb{R}^d$ be the dimension-wise standard deviation vector of the first token before output projection, and let $\mathbf{w}_j \in \mathbb{R}^d$ denote the $j$-th column of $\mathbf{W}_O$ (representing the weights for the $j$-th output neuron). To measure the ordinal association, we compute *Kendall's rank*

*correlation coefficient* ($\tau$) [12] between the absolute weights and the input variance for each output neuron. As shown in Figure 7 (Left), the distribution of $\tau_j$ across all neurons is shifted significantly to the right, with a mean correlation of 0.32. The high degree of structural alignment implies that $\mathbf{W}_O$ consistently assigns larger weights to dimensions where the first token exhibits higher variance.

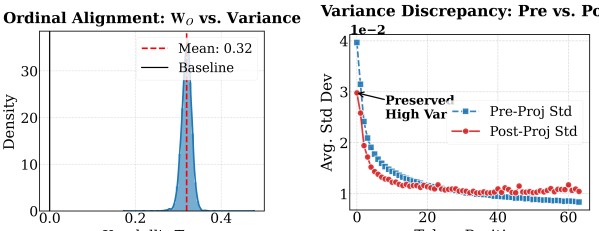

*Figure 7.* **Structural alignment and outlier status preservation in $\mathbf{W}_O$ (Layer 1).** **(Left)** The distribution of rank correlations between $\mathbf{W}_O$ neuron weights and Token 0 input variance. The positive shift (mean=0.32) indicates structural alignment. **(Right)** Even after passing through $\mathbf{W}_O$, the first token maintains significantly higher variance than subsequent tokens.

***Post-Projection Variance Decay*** To verify if this structural alignment effectively preserves the variance discrepancy, we further examine the statistical properties of the output of $\mathbf{W}_O$. Specifically, we compute the dimension-wise standard deviation of the hidden states immediately after the output projection, using random token inputs. As visualized in Figure 7 (right), the *variance decay* pattern persists: the first token retains an exceptionally high variance compared to subsequent tokens. This suggests that the output projection does not drown out the discrepancy; instead, it propagates the first token as a distinct outlier into the residual stream.

### 4.2. Selective Activation of Super Neurons

While $\mathbf{W}_O$ preserves the discrepancy, it does not explain the extreme magnitude of the representation norms observed in Section 3. We hypothesize that the FFN in layer 1 serves as an amplifier to the initial high-variance outlier. To uncover this mechanism, we focus on the gated linear units (GLU) [20] architecture, specifically the SwiGLU variant employed by modern LLMs like Llama-2. The computation is defined in Eq. (2).

***Super Neurons in Weight Matrices*** We first investigate the structural bias in $\mathbf{W}_{gate}$ and $\mathbf{W}_{up}$. We calculate the $l_2$ norms of the weight vectors for each hidden neuron. As shown in Figure 8, specific neurons (e.g., index 7890) possess exceptionally large norms. We term these *super neurons* due to their capacity to capture and amplify signals.

To facilitate a proper structural analysis, we first clarify the *geometric* interpretation of the linear projections involved. Consider a weight matrix $\mathbf{W}$. We interpret the $j$-th column

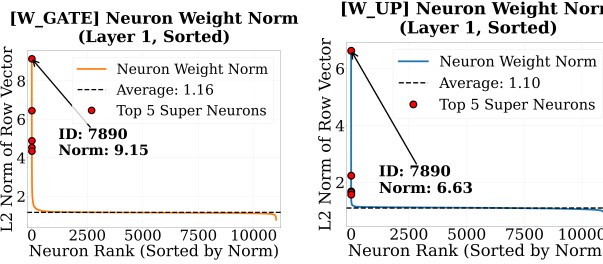

*Figure 8.* **Structural Identification of Super Neurons in Layer 1 FFN.** We visualize the $l_2$ norms of the weight vectors for each hidden neuron in $\mathbf{W}_{gate}$ (**Left**) and $\mathbf{W}_{up}$ (**Right**). A distinct subset of neurons exhibits significantly larger norms.

as the weight vector characterizing the $j$-th neuron, while the $i$-th row represents how the $i$-th dimension of the input activation is mapped to the output space. Based on this formulation, our analysis adopts a dual perspective:

- For the upstream layers ($\mathbf{W}_{gate}$ and $\mathbf{W}_{up}$), we examine their *column vectors*. These correspond to the super neurons that detect and react to the high-variance input of the first token.

- For the downstream layer ($\mathbf{W}_{down}$), we analyze the *row vectors* indexed by the super neurons. These rows determine how the massive activations generated by super neurons are channelled and broadcasted into the residual stream.

With this framework, we trace the propagation of the variance discrepancy through multiple stages in the FFN.

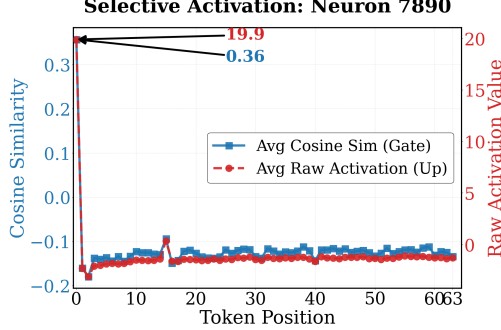

*Figure 9.* **Selective activation of super neuron 7890.** The **Left Axis** shows the cosine similarity with $\mathbf{W}_{gate}^{(7890)}$. The **Right Axis** shows the raw activation via $\mathbf{W}_{up}^{(7890)}$. The first token uniquely achieves both high alignment and massive activation, whereas subsequent tokens are effectively suppressed.

***Selective Activation*** We track the interaction between input tokens and *super neuron* (index 7890) and evaluate whether first token is being treated differently. We measure two metrics for each token position: (1) the cosine similarity between the normalized input token $\mathbf{x}_{norm}$ and the gate

weight column vector $\mathbf{w}_{gate}^{(7890)}$; and (2) the raw activation projected by the up-projection column vector $\mathbf{w}_{up}^{(7890)}$. As visualized in Figure 9, the first token exhibits high positive cosine similarity (opening the gate) and generates massive raw activation. In contrast, subsequent tokens show low alignment or negligible activation. This suggests that the super neuron is selectively triggered by the initial outlier while remaining suppressed for the rest of the sequence.

***Channeling via Sparse Weights*** Finally, we look into how this massive activation $\mathbf{h}_{mid}^{(7890)}$ propagates through the down-projection. We examine the distribution of weights in the row vector $\mathbf{w}_{down}^{(7890)} = \mathbf{W}_{down}[7890, :]$. As shown in Figure 10, the weight distribution is heavy-tailed. Most entries are near zero, but a few specific dimensions exhibit large magnitudes. This channels the massive activation exclusively into these outlier dimensions (e.g., dimension 2533).

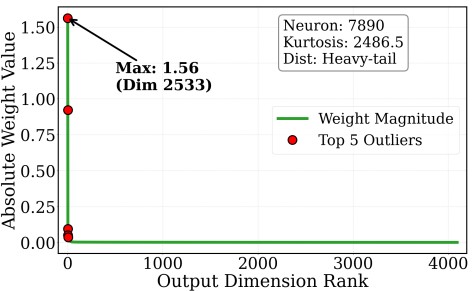

*Figure 10.* **Sparse channeling in down-projection.** The weight distribution corresponding to super neuron in $\mathbf{W}_{down}$ is heavy-tailed. Massive activation is channeled solely into specific outlier dimensions.

### 4.3. Dimension Disparity and Structural Locking in QK Projections

In this section, we analyze the properties of the first token's FFN output, which is generated by massive selective activations of super neurons passing through sparse down-projections. We then investigate how it induces attention sinks in the subsequent self-attention layer.

***Dimension Disparity in the First-Token Representation***
The FFN output introduces significant dimension disparity into the output. Specifically, the representation of the first token becomes dominated by specific outlier dimensions driven by super neuron activations. This disparity compresses the majority of the information into a low-dimensional subspace. To quantify this, we define the *dominance ratio* for the first token's hidden state $\mathbf{h}_0 \in \mathbb{R}^d$. It calculates the ratio of the maximum absolute magnitude to the mean absolute magnitude:

$$\text{DomRatio}(\mathbf{h}_0) = \frac{\max_j |\mathbf{h}_{0,j}|}{\frac{1}{d}\sum_k |\mathbf{h}_{0,k}|}.$$

A higher ratio indicates that the representation is disproportionately concentrated in a few outlier dimensions, reflecting extreme dimension disparity.

As shown in Figure 11, Llama-2 exhibits a sharp rise in the dominance ratio in the shallow layers.

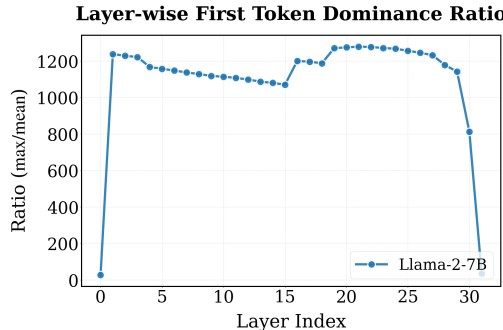

*Figure 11.* **Dimension disparity analysis.** We calculate the layer-wise Dominance Ratio ($\max / \text{mean}$) of the first token on WikiText-2. The sharp rise in early layers suggests that the representation is dominated by a few massive outlier dimensions.

Next, we reveal how this interacts with RMSNorm and the query/key projections in layer 2.

***RMSNorm as Directional Filter*** Let the first token input $\mathbf{x}_0$ be dominated by a massive value $\lambda$ at dimension index $c$ (e.g., index 2533), while other dimensions are negligible. When passing through RMSNorm, the normalization constant is determined almost entirely by $\lambda$. Consequently, the output vector converges to a scaled basis vector $\mathbf{e}_c$:

$$\text{RMSNorm}(\mathbf{x}_0) \approx \text{sgn}(\lambda)\sqrt{d}\gamma_c \cdot \mathbf{e}_c$$

This implies the first token's representation collapses into a fixed direction. We validate this empirically in layer 2. As shown in Table 1, the outlier dimension is orders of magnitude larger than the mean of other dimensions, confirming directional collapse.

*Table 1.* Analysis of the first token's representation after RMSNorm in layer 2. The outlier dimension (index 2533) dominates the vector.

| METRIC | VALUE |
|---|---|
| OUTLIER DIMENSION MAGNITUDE ($|x_{2533}|$) | 1.2568 |
| DIMENSION MEAN ABS VALUE ($|\bar{x}|$) | 0.0048 |
| **DOMINANCE RATIO** | **262.88×** |

***Propagation to Keys*** Consider the key projection in layer 2. Since the normalized input is essentially $\mathbf{e}_c$, the resulting key vector $\mathbf{k}_0^{(h)}$ for the first token in head $h$ approximates the $c$-th row of the key projection matrix $\mathbf{W}_K^{(h)}$:

$$\mathbf{k}_0^{(h)} \approx \pm\sqrt{d} \cdot (\mathbf{W}_K^{(h)})_{c,:} \tag{3}$$

*Alignment across heads*     For an attention sink to form, the score $\langle \mathbf{q}_t^{(h)}, \mathbf{k}_0^{(h)} \rangle$ must be consistently large. We analyze this using two metrics: (1) **structural alignment (SVD)**, which measures the alignment $|\cos(\mathbf{u}_1^{(h)}, \mathbf{k}_0^{(h)})|$ between the sink key and the principal direction of $\mathbf{W}_Q^{(h)}$ extracted via SVD; and (2) **positive ratio**, defined as the proportion of tokens where the dot product is positive. As shown in Figure 12, heads with high structural alignment (tall bars) exhibit a near-100% positive ratio. This indicates that specific heads are structurally predisposed to generate queries that align with the sink key, thereby ensuring large attention scores.

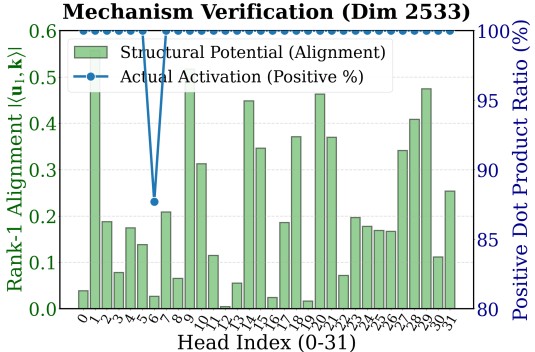

*Figure 12.* **Head-wise analysis in layer 2.** The x-axis represents head indices. The bars (left axis) show structural alignment between the sink key and the query matrix's principal direction. The red line (right axis) shows the ratio of positive attention scores. High alignment correlates with high positivity.

## 5. Practical Implication

Our analysis identifies the structural origin of attention sink: the *variance discrepancy* during value aggregation activates super neurons. The resulting massive activations are channeled through sparse down-projections to create severe *dimension disparity*, which effectively locks the subsequent QK projection and ultimately induces the attention sink.

One reason for variance discrepancy is that standard Softmax function enforces a strict sum-to-one constraint (Eq. (1)). To address this, we can consider replacing Softmax with an unnormalized Sigmoid activation [18]:

$$a_{i,j} = \text{Sigmoid}\left(\frac{\mathbf{q}_i \mathbf{k}_j^T}{\sqrt{d_k}}\right).$$

With Sigmoid attention, the first token is no longer a high-variance outlier, potentially mitigating the formation of attention sinks. This effect is verified in Sec. 5.2.

While replacing Softmax with an unnormalized Sigmoid activation partially mitigates the high-variance first-token outlier, this approach is not ideal: the magnitude and variance of token representations now scale with sequence length, poten-

tially causing training instability. Indeed, prior studies suggest that standard Softmax attention generally exhibits superior training stability and downstream performance compared with alternative unnormalized mechanisms [24, 5, 16]. Consequently, we propose *head-wise RMSNorm*, aiming to retain the standard Softmax formulation and directly address the variance discrepancy.

### 5.1. Head-wise RMSNorm

Recent studies, e.g., DuoAttention [28], have highlighted the functional heterogeneity of attention heads. Motivated by this, we investigate variance discrepancy across different heads and observe a significant *inconsistency*. Specifically, we find that attention heads vary significantly in behavior: some heads are "low-entropy", focusing on few tokens , while other head are "high-entropy", aggregating across a wide range of tokens.

As shown in Figure 13, this leads to a statistical imbalance: low-entropy heads produce high-variance outputs, while high-entropy heads produce low-variance outputs. Without intervention, low-entropy heads dominate the residual stream simply due to their magnitude. Therefore, a **head-wise** intervention is necessary to decouple signal magnitude from attention sparsity.

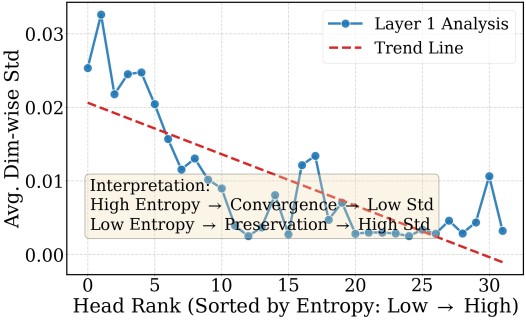

*Figure 13.* **Head imbalance and signal magnitude.** Attention heads are sorted by entropy. Low-entropy heads produce high-variance outputs, while high-entropy heads produce low-variance outputs.

To resolve both the positional variance discrepancy and this head imbalance, we propose *head-wise RMSNorm*, immediately after value aggregation and before the output projection $\mathbf{W}_O$. Specifically, for a head $h$ at position $t$, we normalize the aggregated vector $\mathbf{o}_t^{(h)}$:

$$\hat{\mathbf{o}}_t^{(h)} = \frac{\mathbf{o}_t^{(h)}}{\text{RMS}(\mathbf{o}_t^{(h)})} \odot \boldsymbol{\lambda}, \qquad (4)$$

where $\odot$ denotes element-wise multiplication. Here, $\boldsymbol{\lambda} \in \mathbb{R}^{d_k}$ is a learnable scaling vector shared across all heads ($d_k$ is the head dimension), allowing the model to adaptively recalibrate the feature magnitude for each dimension.

This operation ensures that: (1) **position-wise consistent variance**: the aggregated vectors have a standardized scale regardless of position or context length and, (2) **head-wise consistent variance**: both low-entropy and high-entropy heads contribute equally to the output projection $\mathbf{W}_O$, preventing any single head from acting as a structural outlier.

## 5.2. Experimental Results

**Setup**   We conduct pre-training from scratch on Open-WebText [8] for 40,000 iterations (152M parameters, 20B tokens). We compare three architectures: (1) **Baseline**, the standard Llama-2 architecture using Softmax attention; (2) **Sigmoid attention**, which replaces Softmax with unnormalized Sigmoid [18]; and (3) **Ours (Head-Norm)**, which applies head-wise RMSNorm after value aggregation in standard Softmax attention. All models use the same optimizer (AdamW) and hyperparameters. Details are in Appendix B.

We compare three model variants to investigate the impact of eliminating the *variance discrepancy*. We evaluate the models across three key dimensions: (1) the presence of dimension disparity, (2) the existence of attention sinks, and (3) pre-training convergence speed. Regarding attention sinks, as discussed in Section 1, our method significantly mitigates the phenomenon even without eliminating the root cause of variance discrepancy at its source.

**Alleviation of Dimension Disparity and Manifold Collapse**   We analyze the *dominance ratio* of the first token's hidden states to quantify the severity of dimension disparity. Figure 14 compares the layer-wise trajectory. The **baseline** (red) exhibits a sharp escalation in the dominance ratio starting from early layers, indicating that the first token's representation is effectively hijacked by a single outlier dimension. In contrast, both **Sigmoid** (green) and **ours** (blue) maintain a consistently low dominance ratio. This result provides strong evidence that the extreme dimension disparity is a direct downstream consequence of the variance discrepancy. By eliminating this discrepancy, we successfully disrupt the formation of outliers and preserve a balanced feature distribution.

Consequently, this dimension disparity leads to *manifold collapse*, where the representation is compressed into a low-dimensional subspace. To quantify this, we compute the *effective rank* [19]. Let $p_k = \sigma_k / \sum_j \sigma_j$ be the normalized singular values of the hidden state matrix $\mathbf{H}$. The effective rank is defined as:

$$\text{EffRank}(\mathbf{H}) = \exp\left(-\sum_k p_k \ln p_k\right) \qquad (5)$$

A lower rank indicates that information is concentrated in fewer dimensions. As shown in Figure 15, the baseline exhibits a severe drop in effective rank. In contrast, our

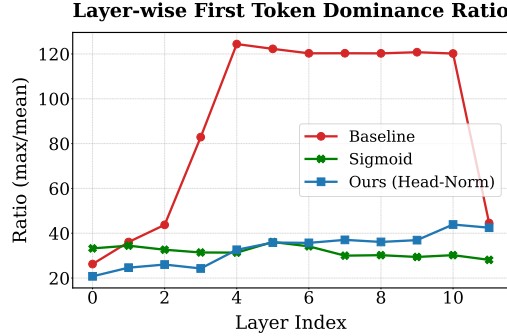

**Layer-wise First Token Dominance Ratio**

*Figure 14.* **Layer-wise dominance ratio of the first token.** The baseline (red) shows a sharp rise, indicating severe dimension disparity where a single feature dominates the representation. Both Sigmoid attention (green) and our method (blue) suppress this dominance.

Head-Norm method maintains a consistently higher effective rank across layers, indicating that resolving the variance discrepancy effectively alleviates manifold collapse and preserves the model's representational capacity.

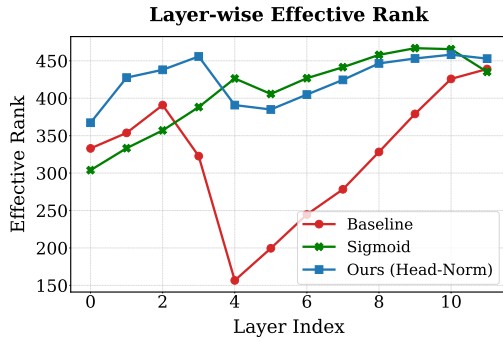

**Layer-wise Effective Rank**

*Figure 15.* **Layer-wise effective rank of the hidden states.** The baseline shows a distinct drop in effective rank, indicating manifold collapse caused by outlier dimensions. Our method maintains a higher effective rank, preserving representational capacity.

**Pre-training Convergence Speed**   We analyze the validation loss trajectories during the pre-training to evaluate optimization efficiency and generalization. As shown in Figure 17, our method (blue) demonstrates significantly faster convergence and achieves lower loss values compared to the baseline (red). In contrast, the unnormalized Sigmoid attention (green) exhibits slower convergence and worse validation performance than the Softmax baseline. This observation highlights that simply replacing Softmax is insufficient; eliminating the variance discrepancy is key to improving the conditioning of the optimization landscape, enabling the model to train more efficiently and generalize better.

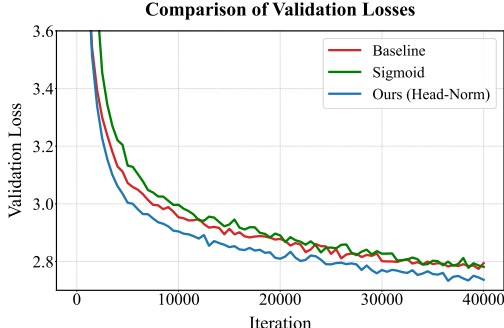

*Figure 16.* Validation Loss

*Figure 17.* **Convergence speed analysis.** Our method (blue) converges significantly faster and achieves lower loss on the validation sets compared to the baseline (red). The standard Sigmoid attention (green) converges slower due to optimization instability.

**Multi-run Consistency**  Table 2 summarizes the results across four pre-training runs with different random seeds. Our HeadNorm method consistently outperforms the baseline across all runs, achieving lower training and validation loss, reduced dimension disparity, and higher effective rank.

*Table 2.* **Evaluation across multiple random seeds.** Results are summarized over four distinct pre-training runs. Mean ± standard deviation are reported. "Layer-wise mean" indicates the value averaged across all transformer layers.

| Metric | Baseline | Ours (HeadNorm) |
|---|---|---|
| Train Loss ($\downarrow$) | $2.7483 \pm 0.0118$ | $\mathbf{2.7073 \pm 0.0095}$ |
| Validation Loss ($\downarrow$) | $2.7812 \pm 0.0109$ | $\mathbf{2.7421 \pm 0.0066}$ |
| Effective Rank (layer-wise mean, $\uparrow$) | $343.71 \pm 15.63$ | $\mathbf{445.96 \pm 5.37}$ |
| Dimension Disparity (layer-wise mean, $\downarrow$) | $82.67 \pm 8.09$ | $\mathbf{33.74 \pm 2.73}$ |

# 6. Discussion

In this work, we provided a mechanistic explanation for the *attention sink* phenomenon. We identified the root cause as a *variance discrepancy* inherently embedded in the causal value aggregation process. Our analysis reveals that the absence of aggregation for the initial token creates a persistent high-variance outlier. This outlier propagates through the network, preserves its magnitude via the output projection ($W_O$), and selectively activates *super neurons* within the FFN. This triggers massive activations that induce severe dimension disparity. These geometric distortions ultimately dominate subsequent Query-Key projections, effectively "locking" the attention mechanism onto the first token.

**Connecting with Prior Explanations**  Prior works often provide a functional perspective on these phenomena. For instance, StreamingLLM [27] attributes attention sinks to the Softmax sum-to-one constraint, suggesting that ini-

tial tokens serve as a repository to store excess attention scores. Similarly, Bondarenko et al. [3] argue that massive activations and attention sinks emerge to help attention heads effectively perform a "no-operation" (no-op). While these studies eloquently explain *how* sinks and outliers are functionally utilized by the model, our contribution is complementary and explores the *upstream* mechanism. We elucidate *why* the sink systematically anchors at position zero in causal decoders. Our causal chain—from variance discrepancy to QK locking—traces the origin of these functional artifacts directly back to the structural asymmetry of causal masking. Furthermore, recent findings by Yona et al. [30] reveal that FFN super-neuron amplification also drives massive activations when processing repeated tokens. Our framework provides a unified structural explanation for this observation: aggregating identical tokens fails to shrink the variance (unlike aggregating diverse tokens). Consequently, repeated tokens structurally mimic the variance behavior of our first-token outlier, triggering the exact same amplification pipeline.

**Mitigation and Structural Implications**  To address the variance discrepancy at its source, we proposed *head-wise RMSNorm* as a variance stabilizer to neutralize statistical outliers before they propagate into the residual stream. Our findings suggest that the empirical success of prior normalization methods may be partially attributed to their unintended mitigation of this underlying variance disparity. Our intervention shares structural similarities with recent works [17, 15], which utilize normalization to address the low-rank bottleneck or to stabilize optimization dynamics on a hypersphere. While the resulting architectures appear similar, our underlying theoretical motivation—restoring variance balance—diverges significantly (detailed in Appendix D).

Beyond addressing attention sinks, our work demonstrates that statistically grounded interventions can effectively mitigate complex geometric anomalies like manifold collapse. While our mechanistic claims are supported by controlled experiments on small-scale models, validating these interventions at a larger scale remains a necessary next step. We hope this foundation encourages the community to further investigate along this path to build inherently more stable and interpretable architectures.

**Limitations and future work**  Our empirical validation was conducted on 152M-parameter models. While attention sinks are known to persist at larger scales, confirming the effectiveness of *head-wise RMSNorm* on models with billions of parameters (e.g., 7B) is a crucial next step. Additionally, future work should investigate how this method interacts with more complex architectures, such as Mixture-of-Experts, to understand its broader applicability and potential limitations in large-scale, heterogeneous settings.

## Impact Statement

This work aims to advance the field of Machine Learning by providing mechanistic insights into attention phenomena and proposing architectural improvements. While there are many potential societal consequences associated with advances in ML, we do not identify any specific ethical or societal risks that warrant detailed discussion in this context.

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

# A. Extended LLMs Analysis

In this section, we provide additional empirical evidence supporting the mechanistic origin of attention sinks. To verify the *universality* of our findings beyond the standard Llama-2 architecture, we conducted identical experiments on *Llama-3-8B*, which utilizes Grouped Query Attention (GQA). We cover the invariant layer-wise onset, the precursor phenomenon of massive representation norms, causal mask interventions, and the direct impact of variance amplification.

## A.1. Invariant Layer-wise Onset and Massive Norms

We first verify the universality of the attention sink onset and its correlation with representation norms on the GQA architecture. Figure 18 shows the same conclusion.

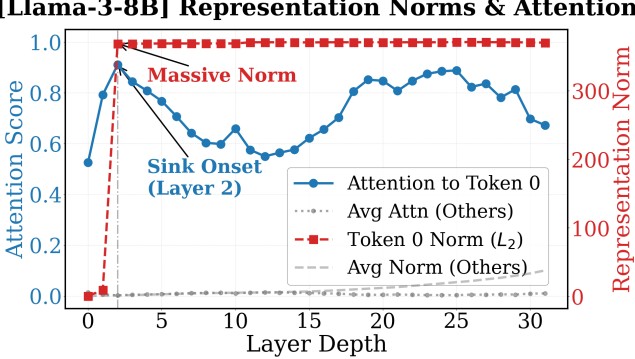

*Figure 18.* **Layer-wise evolution of attention sink and representation norms.** We plot the attention score of the first token (left axis, blue) and its input representation $l_2$-norm (right axis, red) for Llama-3. The synchronized spike indicates that the arrival of a high-norm representation triggers the attention sink.

## A.2. Attention Mask Intervention

To validate that the variance discrepancy is the structural root cause of attention sinks, we intervene on the attention mask of Llama-3. Figure 19 shows the same conclusion.

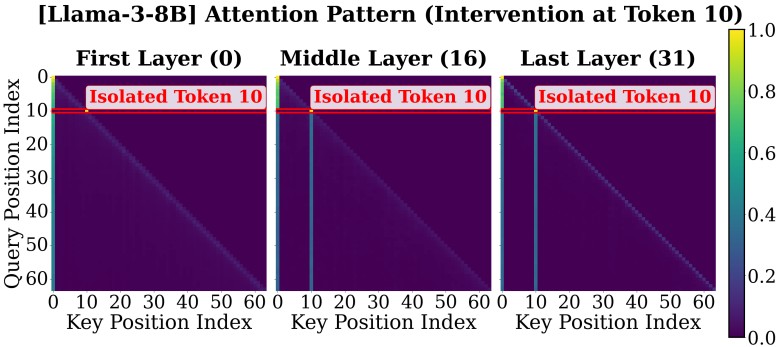

*Figure 19.* **Inducing attention sinks via mask intervention on Llama-3.** We intervene by applying a mask to an arbitrary intermediate token to prevent it from aggregating values (blocking its attention to prior tokens). This intervention effectively induces an attention sink on the targeted non-aggregating token.

## A.3. Direct Variance Amplification Results

We quantitatively demonstrate the causal link between variance and attention scores. We manually amplify the variance of a random token (at index 10) by a factor $\lambda$ and measure the resulting attention score it receives from subsequent tokens.

Figure 20 shows the comparison between the baseline (natural variance, $\lambda = 1$) and the amplified state ($\lambda = 30$) on Llama-3.

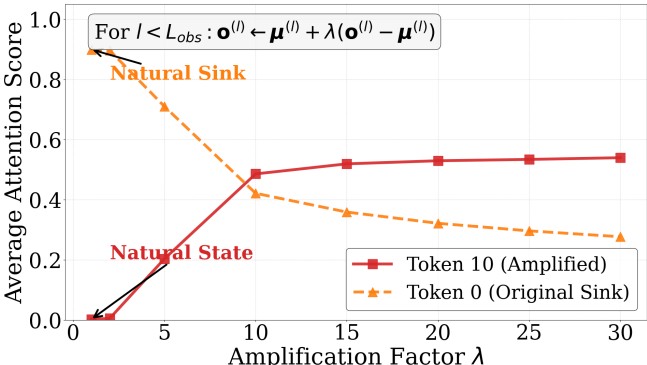

*Figure 20.* **Inducing attention sinks via variance amplification.** We apply a factor $\lambda$ to amplify the variance of an arbitrary token (index 10). Increasing $\lambda$ directly increases the attention score received by the token.

## B. Experimental Setup Details

To ensure reproducibility, we provide the detailed configurations used for the pre-training experiments.

### B.1. Model Architecture

Our baseline models follow the standard **Llama-2** architecture [23], featuring RMSNorm for pre-normalization, SwiGLU activation functions, and Rotary Positional Embeddings (RoPE). The specific architectural hyperparameters for the models used in our main comparisons are listed in Table 3.

*Table 3.* Model Architecture Configurations.

| Hyperparameter | Value |
| --- | --- |
| Hidden Size ($d$) | 768 |
| Intermediate Size ($d_f$) | 3072 |
| Number of Layers ($L$) | 12 |
| Number of Heads ($H$) | 12 |
| Head Dimension ($d_k$) | 64 |
| Vocabulary Size | 50304 |
| Normalization | RMSNorm |
| Activation Function | SwiGLU |
| Position Embedding | RoPE |

### B.2. Initialization and Optimization

Models are initialized using a normal distribution with mean 0. For the majority of layers, including `nn.Embedding` and standard linear projections, we utilize a fixed standard deviation of $\sigma = 0.02$. However, to control variance growth along the residual path, the initialization scale is adjusted specifically for the output mapping layers. Accordingly, the weights for the self-attention output projection ($W_O$) and the FFN downward projection ($W_{down}$) are initialized with $\sigma = \frac{0.02}{\sqrt{2 \cdot L}}$, where $L$ is the number of layers.

Crucially, for our **Head-wise RMSNorm**, we deviate from the standard practice of initializing affine parameters to ones. To prevent the attention output, which now has uniformly scaled variance across all tokens, from abruptly dominating the residual branch and destabilizing training, we initialize the affine parameters **g** to match the **standard deviation of the first token's representation after value aggregation**. This strategy ensures that the output magnitude aligns with the natural, non-decayed scale of the initial token, maintaining stability across the residual connection.

We employ the **AdamW** optimizer with $\beta_1 = 0.9, \beta_2 = 0.95$. The training utilizes a cosine learning rate schedule with a

linear warmup phase. Detailed optimization hyperparameters are provided in Table 4.

## B.3. Infrastructure

All experiments were conducted on a computing node equipped with $8\times$ NVIDIA L40 GPUs using Distributed Data Parallel (DDP) strategy. The total training time was approximately 1.9 days.

## B.4. Dataset

The models are pre-trained on the **OpenWebText** [8] dataset, an open-source reproduction of the WebText dataset. The data is tokenized using the standard GPT-2 tokenizer. Due to the total training volume target, we apply repeated sampling on the dataset during the training process.

*Table 4.* Pre-training Hyperparameters.

| Hyperparameter | Value |
|---|---|
| Peak Learning Rate | 0.001 |
| Min Learning Rate | 0.0001 |
| Warmup Iterations | 2,000 |
| Max Iterations | 40,000 |
| Batch Size | 12 |
| Block Size | 4096 |
| Grad Accumulation Iters | 5 |
| Number of GPUs | 8 |
| Weight Decay | 0.1 |
| Gradient Clipping | 1.0 |
| Precision | bfloat16 |
| Optimizer | AdamW |

## C. Supplementary Empirical Results

### C.1. Consequence of Dimension Disparity: Manifold Collapse

In Section 4.3, we demonstrate that the FFN output exhibits *Dimension Disparity*, where the first token is dominated by specific outlier dimensions. This dominance compresses the representation into a low-dimensional subspace and therefore exhibits *Manifold Collapse*. To quantify this, we compute the *effective rank* [19]. Let $p_k = \sigma_k / \sum_j \sigma_j$ be the normalized singular values of the hidden state matrix $\mathbf{H}$. The effective rank is defined as:

$$\text{EffRank}(\mathbf{H}) = \exp\left(-\sum_k p_k \ln p_k\right) \tag{6}$$

A lower rank indicates that information is concentrated in fewer dimensions.

#### C.1.1. UNIVERSALITY OF MANIFOLD COLLAPSE IN OPEN-SOURSE MODELS

Here, we examine effective ranks of the output of Transformer blocks across layers in two widely adopted open-source models: **Llama-2-7B** and **Llama-3-8B**.

**Experimental Setup:** We feed 20 randomly sampled sequences of length $L = 1024$ from the **WikiText-2** dataset into both models and compute the effective rank of the hidden states output by each Transformer block. The results are averaged over the samples.

**Observations:** As visualized in Figure 21, both Llama-2-7B and Llama-3-8B exhibit a striking similarity in their rank dynamics:

- **Initial Collapse:** We plot the rank trajectory of Llama-2-7B (represented by the **Blue Line**) and Llama-3-8B (represented by the **Orange Line**). Both models start with a relatively high rank at the initial Transformer blocks (Layer 0).

However, immediately after the first few attention and FFN blocks (typically Layers 1–3), the effective rank suffers a sharp, almost vertical decline.

- **Correlation with Outliers:** This collapse perfectly coincides with the amplification of outlier dimensions in the FFN (as discussed in Section 4.2) and leads to the emergence of the attention sink phenomenon. The massive outliers dominate the singular value spectrum, forcing the representation to contract into a low-dimensional subspace.

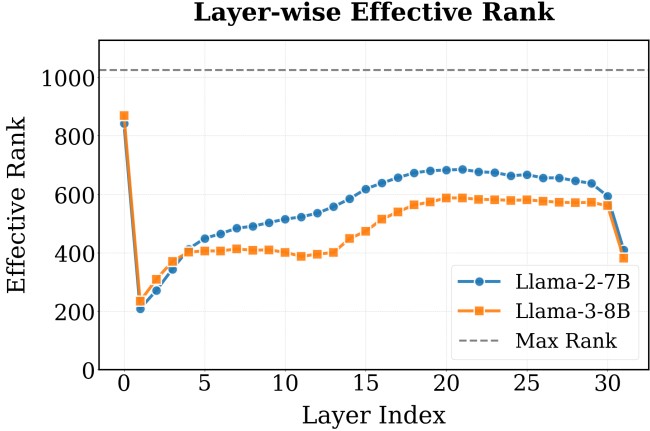

*Figure 21.* **Effective Rank Dynamics in Open-Source Models.** We visualize the effective rank of hidden states across layers. The **Blue Line** represents **Llama-2-7B**, and the **Orange Line** represents **Llama-3-8B**. Both models exhibit a characteristic **precipitous drop** in rank within the first few layers (Shallow Layer Collapse), mirroring the behavior of our baseline. This validates that manifold collapse is a ubiquitous structural anomaly in standard Transformer architectures.

The above results demonstrate that the "shallow-layer manifold collapse" is an intrinsic characteristic of the standard Llama architecture.

## D. More on Related works

To contextualize our findings, we briefly review prior normalization approaches in transformer architectures. While these works did not explicitly target attention sinks, they provide insight into how structural interventions can stabilize representations and improve optimization dynamics.

It's worth noting that [17] introduced a similar normalization step within their Gated Attention mechanism. Their motivation stems from the *low-rank bottleneck* hypothesis: they argue that the composition of two linear projections ($\mathbf{W}_O\mathbf{W}_V$) limits the model's expressivity to a low-rank update. In their view, the normalization acts primarily as a non-linear activation to decouple these linear layers and restore representation rank. Similarly, [15] proposes normalizing all representation vectors onto a hypersphere (using cosine similarity instead of dot product). Their perspective is grounded in *optimization dynamics*, suggesting that constraining features to a unit norm stabilizes the optimization landscape and accelerates convergence.

In contrast to these perspectives, our empirical analysis demonstrates a novel structural role of gating mechanisms in mitigating attention sinks. As shown in Figure 22, both head-wise and element-wise gate scores exhibit a consistent upward trend as the token position advances along the sequence. More importantly, compared to the standard baseline, this progressive increase in gate scores effectively eliminates the dimension-wise variance discrepancy (Figure 23). This provides strong evidence that the gating mechanism directly suppresses the uncontrolled variance amplification of super neurons, thereby resolving the dimension disparity typically observed in early layers. Crucially, this adaptive behavior implies an inherent tendency of the model to eliminate such variance discrepancies among token representations when provided with the necessary structural flexibility.

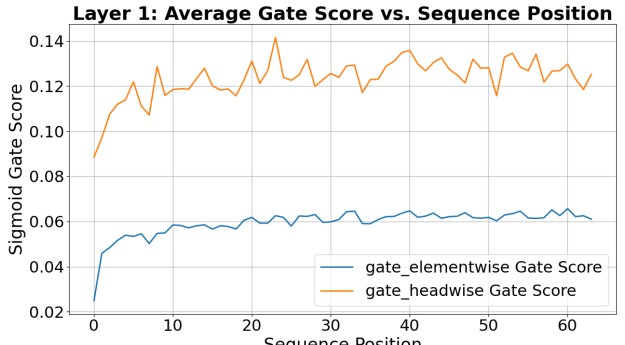

*Figure 22.* Average gate score (head-wise and element-wise) versus sequence position in Layer 1. The gating activation progressively increases along the sequence length.

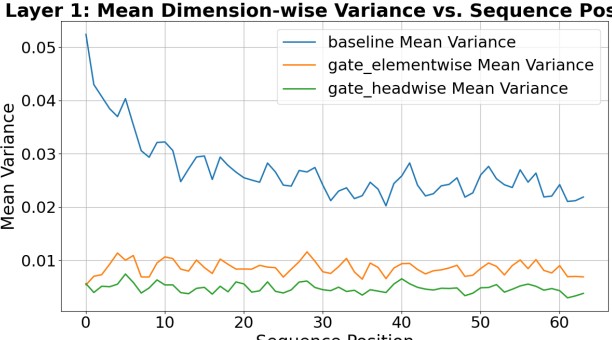

*Figure 23.* Comparison of dimension-wise variance between the baseline and gated models in Layer 1. The application of gating mechanisms successfully eliminates the severe variance discrepancy present in the baseline.

