# OpenReview forum: "The Structural Origin of Attention Sink: Variance Discrepancy, Super Neurons, and Dimension Disparity"
_ICML.cc/2026/Conference — ICML 2026 regular_

### Official Review · Reviewer_5D1n · 2026-03-09

**Soundness:** 3
**Presentation:** 2
**Significance:** 2
**Originality:** 3
**Overall Recommendation:** 4
**Confidence:** 4

**Summary:**

This paper analyzes why decoder-only transformers tend to attend to the first token in a sequence. The authors analyze this behavior from the causal attention mechanism and argue that the asymmetry introduced by causal masking makes the first token have larger variance comparing to others. They then trace how this difference propagates through the transformer component and contributes to sink formation. The paper includes both analytical discussion and empirical experiments to validate the proposed mechanism. Finally, they propose a modification on RMSNorm to mitigate the attention sink phenomenon.

**Compliance With Llm Reviewing Policy:**

Affirmed.

**Final Justification:**

Thanks to the authors for the detailed response and the effort they put into addressing my concerns. All my concerns have been addressed. I think this is a solid paper, though it needs to include more discussion of the related work in the final revision. I am happy to increase my score to 4.

**Key Questions For Authors:**

*   Can the author discuss the difference on the attention weight and the outlier norms pattern in line 121- 152 with existing works [1,2]?

[1] Quantizable Transformers: Removing Outliers by Helping Attention Heads Do Nothing
[2] Yelysei Bondarenko, Markus Nagel, and Tijmen Blankevoort. Understanding and overcoming the challenges of efficient transformer quantization

* Is there any plan to release the code?

**Limitations:**

Section 5 result only conducted on single model. Whether the method generalize to general decoder architecture remain unclear.

**Strengths And Weaknesses:**

### Strengths:

* The paper provides substantial empirical evidence in support of its central hypothesis. The overall empirical study is strong, and the discussion and figures are generally effective in validating the claimed mechanism

* The paper offers a new explanation for why attention sinks emerge.

### Weaknesses:

* Section 5’s empirical evaluation appears to be based on single-run training curves/checkpoints. This is insufficient to support robustness claims about convergence improvements and structural mitigation. At minimum, Figures14, 15, 20, and 21 should be reported over multiple seeds.

* Related-work discussion is too limited. Although the introduction briefly mentions several prior explanations of attention sinks, the paper does not clearly position its variance-based interpretation relative to existing works. This makes it difficult for the reader to understand what is new in the paper and what follows from prior work. For example, how does the proposed variance-discrepancy explanation differ from prior claims that attention tend to implement a no-op, which in turn leads to sink-like behavior? Does the variance-based account complement that explanation, refine it, or conflict with it?

* Same issue as above in appendix. The appendix contains a short related-work discussion, but it only covers two normalization-related papers. A fuller discussion of prior work on attention sinks, no-op behavior, and transformer outliers would make the paper’s contribution much clearer. In particular, prior work has connected sink/outlier behavior to limitations of softmax attention in expressing a true no-op [1,2,4], while this paper explicitly tests sigmoid attention in Section 5 by relaxing softmax’s sum-to-one constrain. [3] is also relevant because it also identifies hidden-state amplification by specific MLP neurons.


### Minor Weakness:
* In Section 3.1, the sentence “We compute the variance for each hidden dimension across the batch and report the average for each position” appears only after the main claim about dimension-wise variance decay. I suggest moving this clarification to the beginning of the section, near the first mention of “dimension-wise variance,” so that readers understand immediately what quantity is being measured.

* There are some overloading of the notation. $k$ sometimes refer to the hidden dimension (section 2) and sometime correspond to the token position (section 3.2). and Eq. (1) uses 1-based summation while the surrounding prose assumes 0-based token indexing.

---

[1] Quantizable Transformers: Removing Outliers by Helping Attention Heads Do Nothing.
[2] Outlier-Efficient Hopfield Layers for Large Transformer-Based Models.
[3] Interpreting the Repeated Token Phenomenon in Large Language Models.
[4] Efficient Streaming Language Models with Attention Sinks.

---

> ### Author Rebuttal · Authors · 2026-03-31
>
> We thank the reviewer for the careful reading and constructive suggestions. The two main concerns appear to be:
> (i) how our variance-based account should be positioned relative to prior work on attention sinks, no-op behavior, and transformer outliers; and
> (ii) how robust the practical results in Section 5 are beyond a single training setup.
> We address these first, and then respond to the presentation comments and code-release question.
>
> ### **Positioning our contribution relative to prior sink / no-op / outlier literature**
>
> We agree that the related-work discussion should be expanded. Prior works such as **Quantizable Transformers**, the transformer quantization work cited as [2], and **StreamingLLM** mainly provide a **functional** account: sink/outlier behavior arises because softmax attention cannot easily implement a true no-op, so residual probability mass is routed to certain tokens. Our contribution is complementary and more **mechanistic/upstream**: we explain why the sink systematically appears at **position 0** in causal decoders.
>
> Specifically, our paper supports the chain
>
> $\text{causal value aggregation} \rightarrow \text{variance discrepancy} \rightarrow W_O \text{ preservation} \rightarrow \text{FFN super-neuron amplification} \rightarrow \text{dimension disparity} \rightarrow \text{QK locking / sink}.$
>
> Thus, prior work explains how sinks are **functionally used**, while we explain how they are **structurally created**. For lines 121–152, we do not merely observe the attention/norm correlation, but trace its origin to causal masking asymmetry.
>
> Paper [3] identifies MLP-neuron amplification in repeated-token behavior. Sec. 4.2 agrees but identifies the underlying structural cause: **we believe this is** **because aggregating identical tokens does not shrink variance (unlike diverse text), their repeated tokens structurally mimic our first-token outlier.**
>
> ### **Robustness of Section 5**
>
> We agree that multi-seed reporting for Figs. 14, 15, 20, and 21 would strengthen Section 5. At the same time, the paper’s **main mechanistic claims do not rely on a single training curve**: they are established in Secs. 3–4 by the variance measurement, mask intervention, centered-variance amplification, and FFN/QK analyses, and Appendix A already shows the same onset / mask / amplification behavior on **Llama-3-8B (GQA)**.
>
> To address generalization more directly, we additionally ran HeadNorm on larger Llama-2 variants (n_layer=24, n_head=16, n_embed=1024) and on a standard GPT-style decoder. **Collectively, all experiments consistently support our core conclusions across diverse architectures.**
>
> For Llama-2, we observe reduced first-token [dimension disparity](https://anonymous.4open.science/r/ICML2026-rebuttal-B7E0/comparison_of_dimension_disparity_mediumsize_llama2.png) and [sink strength](https://anonymous.4open.science/r/ICML2026-rebuttal-B7E0/comparison_of_attention_sink_mediumsize_llama2.png), higher [effective rank](https://anonymous.4open.science/r/ICML2026-rebuttal-B7E0/comparison_of_effective_rank_mediumsize_llama2.png), and faster [training](https://anonymous.4open.science/r/ICML2026-rebuttal-B7E0/comparison_of_training_losses(mediumsize_llama2).png) / [validation](https://anonymous.4open.science/r/ICML2026-rebuttal-B7E0/comparison_of_validation(mediumsize_llama2).png) convergence.
>
> The GPT-style decoder exhibits the exact same trends: reduced [dimension disparity](https://anonymous.4open.science/r/ICML2026-rebuttal-B7E0/comparison_of_dimension_disparity_smallsize_gpt.png) and [sink strength](https://anonymous.4open.science/r/ICML2026-rebuttal-B7E0/comparison_of_attention_sink_smallsize_gpt.png), higher [effective rank](https://anonymous.4open.science/r/ICML2026-rebuttal-B7E0/comparison_of_effective_rank_smallsize_gpt.png), and faster [training](https://anonymous.4open.science/r/ICML2026-rebuttal-B7E0/comparison_of_training_losses(smallsize_gpt).png) / [validation](https://anonymous.4open.science/r/ICML2026-rebuttal-B7E0/comparison_of_validation(smallsize_gpt).png) convergence. We will add these results in the revision, and also report multi-seed curves for the original setup.
>
> ### **Minor presentation fixes**
>
> We will implement all presentation suggestions: moving the Sec. 3.1 variance statistic clarification to the beginning, standardizing notation to avoid overloading token/dimension indices, and making Eq. (1) consistent with 0-based indexing. These clarify without changing conclusions.
>
> ### **Code release**
>
> Yes, we will release the code for our analysis, interventions, and HeadNorm training. Overall, the requested revisions clarify positioning and robustness without changing our core established mechanisms.

---

> > ### Author Rebuttal · Reviewer_5D1n · 2026-04-03
> >
> > Thanks for the time to answer my concerns. I have some follow up questions.
> >
> > * For related work, thanks for the clarification. I now understand the paper’s position better. I suggest the authors broaden the related-work discussion and include a dedicated section that positions the paper against prior analyses of attention sinks, no-op behavior, and related outlier phenomena. This section may go in the main text or the appendix.
> >
> > * For robustness, thanks for the additional results. They show that the effect is not tied to one architecture. However, I still do not understand why the extra experimental budget went to larger or different architectures instead of multi-seed runs for the original Section 5 setup. Even across architectures, single-run curves may still reflect favorable seed selection. I apologize if this sounds too direct, but I think this point matters for robustness in an ICML paper. I'll appreciate clarification on the experiments choice. I may have missed something.

---

> > > ### Author Response · Authors · 2026-04-06
> > >
> > > We thank the reviewer for the prompt follow-up. We address the two points below.
> > >
> > > **Related Work**
> > > We agree with the suggestion. In the revision, we will include a section to explicitly position our mechanistic explanation against prior analyses of attention sinks, no-op behaviors, and related outlier phenomena.
> > >
> > > **Choice of Experiments and Multi-Seed Robustness**
> > >
> > > In the first rebuttal, we interpreted your original review as raising two separate issues: robustness to seed choice in Section 5, and whether the effect generalizes beyond a single decoder configuration. Because the original paper only reported one training setup, and your review explicitly questioned architectural generality, we first used the limited rebuttal-time budget to check breadth: a larger Llama-2 variant and a standard GPT-style decoder. In hindsight, that breadth evidence is complementary, but it is not a substitute for the direct multi-seed test on the original Section 5 setting.
> > >
> > > We agree that multi-seed runs are essential to rule out seed selection bias. Following your follow-up, we have completed additional pre-training runs using the exact Section 5 setup with three new random seeds (0, 1000, and 2000). The metrics below summarize the results across all four distinct runs (including the original experiment from the submitted paper):
> > >
> > > | **Metric** | **Baseline** | **HeadNorm (Ours)** |
> > > | --- | --- | --- |
> > > | **Train Loss** ($\downarrow$) | $2.7483 \pm 0.0118$ | $\mathbf{2.7073 \pm 0.0095}$ |
> > > | **Validation Loss** ($\downarrow$) | $2.7812 \pm 0.0109$ | $\mathbf{2.7421 \pm 0.0066}$ |
> > > | **Effective Rank** (layer-wise mean, $\uparrow$) | $343.71 \pm 15.63$ | $\mathbf{445.96 \pm 5.37}$ |
> > > | **Dimension Disparity** (layer-wise mean, $\downarrow$) | $82.67 \pm 8.09$ | $\mathbf{33.74 \pm 2.73}$ |
> > >
> > > These are mean ± std over 4 seeds.  Importantly, the gaps between the baseline and our HeadNorm are significantly larger than the across-seed std on all four quantities, which directly addresses the concern that the original Section 5 result might reflect a favorable single seed. This table is meant to cover the quantities underlying Figs. 14, 15, 20, and 21. In the revision, we will update Section 5 to report the original setting with multi-seed statistics, while keeping the cross-architecture results as complementary evidence of generalization.

---

### Official Review · Reviewer_tSoo · 2026-03-12

**Soundness:** 3
**Presentation:** 2
**Significance:** 3
**Originality:** 3
**Overall Recommendation:** 5
**Confidence:** 4

**Summary:**

This paper offers a mechanistic narrative for attention sinks that nicely collect many terms (causal masking, null attention, super neurons) tying together (i) a position-dependent variance decay induced by causal value aggregation, (ii) propagation through ​output-projection of attention mech., (iii) selective FFN super neuron activation and sparse down-projection, and (iv) a geometric view of how RMSNorm turns dimension disparity into QK locking and a sink in the next layer. The synchronization between the first-token norm spike and sink onset (Fig. 3) is a strong motivating observation, and the later decomposition into structural stages is clear and useful.

**Compliance With Llm Reviewing Policy:**

Affirmed.

**Final Justification:**

Initially I was not fully convinced by the proposed causal chain induced by variance, but the authors clarification made it clearer with new experiments on how the variance pattern induced by causal value aggregation is carried through the residual stream (for instance level) and then selectively amplified by FFN super neurons producing dimension disparity that RMSNorm can convert into a stable directional bias which ultimately results in attention sink. With this clarified, I find the paper’s mechanistic decomposition both insightful and useful, and the interventions provide meaningful evidence for the proposed story.

**Key Questions For Authors:**

I would recommend the authors add one or two sentences clarifying:

* In the amplification of variance experiment, does variance here correlate with range?
* what inputs are used for Fig. 9 (random tokens vs WikiText-2 vs something else),
* whether the cosine similarities / raw activation are averaged across a set of sequences (and how many), analogous to Sec. 3.1.
* In the variance-amplification intervention increasing \lambda also increases the empirical magnitude of representation (not just its variance). How do you disentangle magnitude from overall spread as the driver of the sink? My concern is that increasing (\lambda) appears to do more than increase the second moment across sequences it also makes token (k) an outlier in magnitude. Intuitively, that alone could make the token more attractive to attention downstream (e.g., by inducing larger activation or larger QK scores).  Also what exact statistic do you mean by \mu in this experiment.

**Limitations:**

yes

**Strengths And Weaknesses:**

## Strengths
The paper nicely draw a mechanstic pipeline to answer how attention sink emerge : it argues output-projection preserves the outlier status but does not explain the extreme magnitudes, and then propose that the layer-1 FFN is the actual amplification stage. This is then supported by the structural identification of super neurons via unusually large column norms of the upper stream projections (gate and input) in the FFN (equation 2), and by the observation that one (or few) such neuron is selectively activated by token 0 while later tokens are suppressed (see questions). The follow-up that the corresponding down-projection' row is heavy-tailed, channeling the activation into a few dimensions, gives a plausible bridge to the observed dimension disparity and subsequent sinks. Followed bu successful practical utilization of these insights.

* Overall, I find the mechanistic decomposition valuable and the proposed mitigation direction (statistical parity restoration) well motivated,
* but I would make my positive assessment conditional on clarifying the input distribution and averaging protocol used for the Sec. 4.2 (see below).
 ## Comments
 I felt the paper left me with unanswered question in the end, mainly how the variance (which is measured for a set of samples) can still explain the attention sink on each given instant from this set.

my main concern is that the causal story “high-variance outlier → selective super neuron activation → huge norm → sink” is only fully convincing if the Sec. 4.2 measurements are clearly computed under an input distribution aligned with the variance experiments in Sec. 3.1.
In Sec. 3.1, the paper is very explicit: to avoid BOS-related zero variance, they use fully random token sequences, compute per-dimension variance across the batch, and report position-wise averages. However, in Sec. 4.2 (Fig. 9), when reporting cosine similarity between the normalized input token normalized inputs (x) and super neuron columns weights, the text does not state what dataset or input distribution is used, nor whether the curves are averaged across many sequences batches or shown for a single batch.

This matters because, as written, the selective alignment result could be interpreted as either (a) a robust, distributional phenomenon tightly tied to the variance mechanism, or (b) a behavior that depends on a particular dataset, prompts, or batch realization. Since the paper’s core claim is that sinks persist even for random tokens and are structurally induced, tightening this experimental specification would make the mechanistic chain much cleaner.


## Minor comments:
* In the RMSNorm directional-collapse approximation what is \gamma,
* Figure 3 legend doesn’t show Token 0, legend box can move left or rights.
* Figure 3 , attention to token 0 is computed from what position!, should be clarified.
* Several equations are not referenced.

---

> ### Author Rebuttal · Authors · 2026-03-31
>
> We thank the reviewer for identifying this important interpretation point. Below, we clarify the link between Sec. 3.1's distribution-level variance and instance-level sinks, and detail our experimental protocols.
>
> ### **Distribution-level variance vs. instance-level sink**
>
> Our claim is **not** that a batch-level variance number directly determines the attention score in a single forward pass. Rather, Sec. 3.1 establishes a **distributional positional bias** created by causal value aggregation. Because position 0 is the only position that is not averaged, while later positions are convex averages of more value vectors, the representation distribution at position 0 retains a larger **centered spread around the value-space mean** than later positions. In this sense, position 0 is structurally much more likely to realize an outlier.
>
> Once realized, the subsequent mechanism is entirely **instance-level**. As mapped in our paper: this outlier selectively activates super neurons (Fig. 9), channels sparsely through $W_{\text{down}}$ (Fig. 10), and causes dimension disparity leading to QK locking (Fig. 11).
>
> So the role of Sec. 3.1 is to explain **why token 0 is systematically predisposed to become the outlier**, not to claim that a variance scalar by itself determines every individual attention map. We will make this bridge more explicit in the revision, but the mechanism itself is already present in the current submission.
>
> ### **Experimental specification for Sec. 4.2 / Fig. 9**
>
> We agree that the protocol for Sec. 4.2 / Fig. 9 should be stated more explicitly. In our experiments, **Fig. 9 is computed on WikiText-2, and both the cosine-similarity curve and the raw-activation curve are averaged position-wise over 500 sequences**. Most of the mechanistic analyses in the paper are on real text; random tokens are used only in the specific places where they are necessary to expose the variance effect cleanly. **This is justified since Sec. 3 shows random and real sequences exhibit identical sink representations.**
>
> Sec. 3.1 uses random tokens because a fixed BOS token causes degenerate across-batch variance, hiding position 0's outlierness. Thus, the use of random tokens in Fig. 4 is a measurement choice, not a change in mechanism.
>
> To directly align Sec. 4.2 with Sec. 3.1, we additionally reran the Fig. 9 / Fig. 11 analyses under the same random-token setup as Sec. 3.1, again averaging over 500 sequences. The same qualitative pattern remains ([Fig. 9](https://anonymous.4open.science/r/ICML2026-rebuttal-B7E0/random_sequences_input_for_fig9.png), [Fig. 11](https://anonymous.4open.science/r/ICML2026-rebuttal-B7E0/random_sequences_input_for_fig11.png)): token 0 aligns with super neurons, exhibits massive activation, and concentrates dimension disparity. This strengthens the same structural story already argued in the paper. We will add clarification and include the random-input version in the revision.
>
> ### **Clarifying Fig. 6: centered-variance amplification, $\mu$, and why the effect is not simply norm scaling**
>
> We agree that the phrase “variance amplification” can be read too loosely. In Fig. 6, the manipulated quantity is better described as **centered-variance amplification**:
>
> $o_k^{\prime(l)}=\mu^{(l)}+\lambda\bigl(o_k^{(l)}-\mu^{(l)}\bigr)$,
>
> where $\mu^{(l)}$ is the layer-l mean value vector, estimated under the random-token distribution by averaging over batch and sequence positions (**50k sampled tokens in total**). Thus, the experiment does **not** simply multiply the token’s norm; it increases the token’s **centered deviation from the value-space mean**. For this reason, in the revision we will refer to this experiment more precisely as **centered-variance amplification**.
>
> Addressing concerns on magnitude vs. spread: our intervention around $\mu^{(l)}$ proves the mechanism relies on value-distribution outlierness, not raw norm. We explicitly demonstrate this with a new direct control:$\text{centered scaling: } o'=\mu+\lambda(o-\mu)
> \qquad\text{vs.}\qquad
> \text{pure magnitude scaling: } o'=\lambda o.$
>
> Empirically, pure magnitude scaling has little effect on inducing a sink, whereas centered scaling reliably induces one ([Fig. 6 control](https://anonymous.4open.science/r/ICML2026-rebuttal-B7E0/norm_for_fig6.png)). Thus, the causal driver in Fig. 6 is the token's centered deviation from $\mu$, rather than absolute magnitude or coordinate-wise range.
>
> This intervention bridges our paper's two levels: Sec. 3.1 identifies token 0's distributional tendency to deviate, while Fig. 6 shows artificially recreating this outlierness induces the downstream sink.
>
> ### **Minor clarifications**
>
> We will also incorporate the requested presentation fixes: define $\gamma$**(the affine parameter)** explicitly in the RMSNorm approximation, clarify Fig. 3's query-averaged attention, improve legends, and fix equation references.

---

> > ### Author Rebuttal · Reviewer_tSoo · 2026-04-02
> >
> > Thanks to the authors for the clarifications and additional experiments. They addressed my main concerns, and I now have a clearer understanding of the causal chain proposed in the paper. Based on this Im increasing my score from 3 to 5.

---

> > > ### Author Response · Authors · 2026-04-06
> > >
> > > We are truly grateful to the reviewer for the increased score and for the recognition that our rebuttal has fully addressed the initial concerns. We are committed to incorporating all new results and clarifications discussed during this period into the final manuscript. Thank you for this invaluable and constructive dialogue.

---

### Official Review · Reviewer_6Xox · 2026-03-13

**Soundness:** 3
**Presentation:** 3
**Significance:** 3
**Originality:** 3
**Overall Recommendation:** 4
**Confidence:** 5

**Summary:**

The paper hypothesizes that attention sinks in Llama-type architectures originate from a variance discrepancy in the value aggregation, itself directly coming from the way attention is masked, leaving the first token untouched. Through a set of empirical investigations, the authors propose interventions shedding light on such responsible phenomena and finds way to alleviate it in two ways: (I) changing the mask (ii) artificially removing this variance discrepancy. Beyond these mechanistic explanations, the authors propose a normalization that should mitigate attention sinks.

**Compliance With Llm Reviewing Policy:**

Affirmed.

**Key Questions For Authors:**

Questions:
- Could the authors explain further on why there is a need to use random tokens in their figure 4. I appreciate that the BOS token would lead to a null-variance but this seems to contradict current observations of attention sinks emerging on the BOS token specifically. Does it mean that your hypothesis that the variance discrepancy is responsible for attention sinks is not the root cause of attention sinks observed in  practice? Could the authors comment on this important point.
- In figure 5, could you show what happens at layer 11 instead? It seems like the most interesting layer to look into if we want to replicate the observations at layers 1 or 2 for the BoS token. How about generating the plot of Figure 4 for layer 11 as well?
- In figure 6, is the attained value of 0.5 because the 1st token has already taken 50% of the attention score? Could you add a line on the plot showing the attention score of the first token as well?
- Line 656: can you show what is the resulting values for lambda as the model is trained?
- I wonder how Figure 14 would look like with PostNorm instead of PreNorm. Moreover, could figure 15 be plotted across 3 runs?

**Limitations:**

yes

**Strengths And Weaknesses:**

Strengths:
- The paper is very well written, polished, well structured and easy to follow.
- The topic under study is of high practical relevance with potential high impact on the field.
- The empirical validations are strong, the ablations are thorough and performed on sufficiently large models.

Weaknesses:
- As it is, it is not so clear to me how these findings confirm or invalidate previous analyses already done in the literature on attention sinks (especially given how flourished the literature can be).
- A big portion of the ablations are done on random tokens, would be good to see how these observations translate when handling textual real world data.

---

> ### Author Rebuttal · Authors · 2026-03-31
>
> We thank the reviewer for the careful reading and constructive suggestions. We believe the two main concerns are: (i) how our findings should be positioned relative to prior work on attention sinks, and (ii) why some of our mechanistic ablations use random tokens. We address these first, and then respond point-by-point to the specific questions.
>
> ### **Positioning relative to prior work**
>
> Our goal is not to invalidate previous analyses, but to add a different level of explanation. Prior work has mostly characterized what attention sinks *do* once they are present—for example, as a recipient of residual attention mass, a mechanism related to mixing control, or a phenomenon associated with positional/spectral structure. Our contribution is a mechanistic account of *how* such a sink is created: causal value aggregation creates a position-dependent outlier, amplified in the FFN through super neurons, converted into dimension disparity, and then induces QK locking. The mask intervention and variance-amplification intervention support this claim because they produce sinks at arbitrary positions without relying on token semantics. We will revise the related-work discussion to make this complementary relationship explicit.
>
> ### **Why random tokens appear in some ablations**
>
> Most of our observational results are on real text. Random tokens are used only in a small number of controlled diagnostics whose purpose is to isolate the structural effect from token-identity confounds. Specifically, Fig. 4 measures across-batch variance. A fixed BOS token at position 0 has zero across-batch variance, hiding the outlier. We randomize the first token to make this statistic work. In practice, the sink targets BOS simply because it always occupies this uniquely unaggregated position 0.
>
> To make this clearer, we have added new figures detailing the model's behavior when taking random sequence as input. As shown in the new figures ([Fig. 9](https://anonymous.4open.science/r/ICML2026-rebuttal-B7E0/random_sequences_input_for_fig9.png), [Fig. 11](https://anonymous.4open.science/r/ICML2026-rebuttal-B7E0/random_sequences_input_for_fig11.png)), the entire causal chain of the attention sink—specifically the selective activation of super neurons and the resulting dimension disparity—is identical to the phenomena observed under real text.
>
> **Q2: Fig. 5 and Fig. 4 plot for layer 11.**
>
> We are not quite sure why layer 11 is specifically mentioned (typo?). We will provide the Layer 11 results in our next response. In the meantime, we analyzed the [Layer 2 attention map](https://anonymous.4open.science/r/ICML2026-rebuttal-B7E0/layer2_for_fig5.png) and its corresponding [variance statistic](https://anonymous.4open.science/r/ICML2026-rebuttal-B7E0/layer2_for_fig4.png) as they mark the initial onset of the sink. The results show that dimension disparity restricts position-0 representations to specific $W_V$ rows, leading to low positional variance and once the sink is established, the variance across the sequence stabilizes. This suggests a direct causal link: **the attention sink acts as a structural stabilizer for representation variance.**
>
> **Q3: Is the $\approx 0.5$ value in Fig. 6 because the first token already holds about half the attention mass?**
>
> Yes. The plateau is consistent with competition between the original position-0 sink and the amplified token. Following the reviewer’s suggestion, we now add a [second curve](https://anonymous.4open.science/r/ICML2026-rebuttal-B7E0/token0_for_fig6.png) showing the attention assigned to token 0. As $\lambda$ increases, the amplified token gains attention while the original sink loses it, making this competition explicit.
>
> **Q4: What values does the learned** $\lambda$ **take during training?**
>
> This is an important diagnostic of HeadNorm. We tracked the learned affine parameter ($\lambda$) across training iterations. Empirically, the global mean of $\lambda$ steadily decreases as training progresses ([trajectory of global $\lambda$ mean](https://anonymous.4open.science/r/ICML2026-rebuttal-B7E0/learned_gamma_llama_smallsize.png)). This supports the validity of our HeadNorm.
>
> **Q5: PostNorm and 3-run robustness for Fig. 15.**
> Our experiments focus on PreNorm because it is standard for modern LLMs. During the rebuttal window, we prioritized cross-scale and cross-architecture training runs, which already show the same qualitative improvements on larger Llama-2 models and a standard GPT-style decoder. Specifically, we added new pre-training experiments on a [medium-sized Llama-2](https://anonymous.4open.science/r/ICML2026-rebuttal-B7E0/comparison_of_training_losses(mediumsize_llama2).png)($n_{layer}=24, n_{head}=16, n_{embed}=1024$) and a [small-sized GPT](https://anonymous.4open.science/r/ICML2026-rebuttal-B7E0/comparison_of_training_losses(smallsize_gpt).png). We agree that multi-seed curves would strengthen Section 5, and a PostNorm comparison will also be in interesting complementary study.

---

> > ### Author Rebuttal · Reviewer_6Xox · 2026-04-04
> >
> > I thank the authors for their engagement with all reviewers.
> >
> > Overall, I find this work interesting and I maintain my positive assessment of this work. (Regarding Q2, indeed this was a typo, thanks for noting it).

---

> > > ### Author Response · Authors · 2026-04-06
> > >
> > > Thank you for the thoughtful follow-up and for confirming that our rebuttal resolved your concerns. We will incorporate the clarified positioning relative to prior attention-sink work, the rationale for the random-token diagnostics, and the additional plots and diagnostics discussed during rebuttal into the final revision. We appreciate your careful reading and constructive suggestions.

---

### Official Review · Reviewer_GZnF · 2026-03-22

**Soundness:** 3
**Presentation:** 4
**Significance:** 3
**Originality:** 4
**Overall Recommendation:** 4
**Confidence:** 4

**Summary:**

This paper aims to explain the cause of attention sink by the variance discrepancy of tokens at different positions in the value aggregation process. This thought is validated by i) changing the causal mask causes changes in attention sink position, ii) changing the norm (variance) of the token will change the attention sink position. Finally, a head-wise RMSNorm method is proposed to address this issue.

**Compliance With Llm Reviewing Policy:**

Affirmed.

**Final Justification:**

Overall, it is a valuable work, but I am still a bit concerned about the correctness and efficiency in more general cases, so I will keep my rating.

**Key Questions For Authors:**

Is there any harmness of the proposed RMSNorm to the expressability of the model?

**Limitations:**

yes

**Strengths And Weaknesses:**

Strengths:
- The paper is written well in a good format. All the figures and tables are very clear.
- The thought chain of the reason for the attention sink is very sound and persuasive.
- Thought experiments, including changing the causal mask position and changing the norm of tokens, are well designed and able to prove the explanation is correct.

Weaknesses:
- The anaslysis are all static with pre-trained weights. It is better to show if it holds in random initialization and the dynamics in training.
- The proposed solution is only tested in very simple toy experiments. It's better to do more tests to show its effectiveness and also check it in training.

---

> ### Author Rebuttal · Authors · 2026-03-31
>
> We thank the reviewer for the positive assessment. We address the concerns regarding the static nature and scale of our experiments below.
>
> ### **Static nature of our analysis on attention sinks**
>
> This work aim to uncover the structural orgin of attention sinks. Naturally, we started by investigating pre-trained LLMs that already exhibits the sink behaviors and revealed their inherent mechanism.
>
> Nevertheless, we agree that extending our investigation to models at random initialization and during actual training dynamics broadens the impact of our findings. To this end, we conducted several new experiments.
>
> **1. Variance Discrepancy at Random Initialization**
> We analyze the model at pure random initialization and simulate the input to the self-attention module (post pre-attention norm) using a standard normal distribution (mean=0, variance=1).
> As illustrated in the newly added figure ([$1/\sqrt{t}$ variance decay](https://anonymous.4open.science/r/ICML2026-rebuttal-B7E0/random_init_variance_discrepancy(smallsize_llama2).png)), the standard deviation of aggregated values in a randomly initialized model decays proportionally to $1/\sqrt{t}$ (where $t$ denotes the sequence position). This demonstrates that the variance imbalance is inherent to the architecture at initialization, naturally setting the stage for the attention sink to emerge during training.
>
> **2. Larger scale experiments and tracking evolvement during training**
> Regarding the concern about "toy experiments," we expand our empirical validations to larger-scale models within the Llama-2 family(n_layer=24，n_head=16，n_embed=1024) and generalize our HeadNorm solution to standard GPT-style architectures.
>
> During the training process, we track the evolvement **dimemsional disparity** and **attention sink** for both the baseline model and model with our HeadNorm modification. The findings are:
>
> - It continuously suppresses dimension disparity (measured by dominance ratio in lines 321-324) from initialization throughout the entire training process, as verified in both [small-sized](https://anonymous.4open.science/r/ICML2026-rebuttal-B7E0/training_dynamics_of_dimension_disparity(smallsize_llama2).png) and [medium-sized](https://anonymous.4open.science/r/ICML2026-rebuttal-B7E0/training_dynamics_of_dimension_disparity(mediumsize_llama2).png) Llama-2 variants.
> - It maintains a significantly lower attention sink throughout training, as demonstrated in both [small-sized](https://anonymous.4open.science/r/ICML2026-rebuttal-B7E0/training_dynamics_of_attention_sink(smallsize_llama2).png) and [medium-sized](https://anonymous.4open.science/r/ICML2026-rebuttal-B7E0/training_dynamics_of_attention_sink(mediumsize_llama2).png) Llama-2 variants.
> - Most importantly, resolving this structural variance discrepancy leads to noticeably faster training convergence, as demonstrated in both [medium-sized Llama-2](https://anonymous.4open.science/r/ICML2026-rebuttal-B7E0/comparison_of_training_losses(mediumsize_llama2).png) and [small-sized GPT](https://anonymous.4open.science/r/ICML2026-rebuttal-B7E0/comparison_of_training_losses(smallsize_gpt).png) models.
>
> These added experiments further support the validity of our findings about the sink mechanism and the proposed HeadNorm as a intervention method. We will add this part to the appendix.
>
> ### **Impact of RMSNorm on Model Expressability**
>
> We thank the reviewer for this important question. Strictly speaking, HeadNorm is not fully expressivity-preserving relative to vanilla attention, since normalizing each head’s aggregated output removes its absolute norm, i.e., one scalar degree of freedom per head/token.
>
> However, we believe this restriction is targeted rather than harmful. HeadNorm still preserves the relative feature pattern within each head up to a learnable per-dimension rescaling, and the subsequent output projection $W_O$ still provides flexible cross-head mixing. In our setting, the quantity being removed is precisely the unstable magnitude component induced by variance discrepancy, which causes certain tokens/heads to dominate due to norm rather than content.
>
>  Empirically, we do not observe reduced usable capacity. On the contrary, HeadNorm suppresses the first-token dimension disparity (Fig. 14), maintains a consistently higher layer-wise effective rank than the baseline (Fig. 20), and improves both training and validation convergence (Figs. 15 and 21).
>
> We view the effective-rank result as the most relevant evidence for expressivity: it shows that the hidden states are less collapsed and occupy a broader subspace, instead of being dominated by a few outlier coordinates. We emphasize that faster convergence is primarily an optimization signal rather than a direct expressivity metric, but it further indicates that HeadNorm does not introduce a practical capacity bottleneck. Overall, our current evidence suggests a mild restriction in raw radial freedom, but a net gain in effective representational capacity.

---

> > ### Author Rebuttal · Reviewer_GZnF · 2026-04-06
> >
> > Thanks for the rebuttal. Overall, it is a valuable work, but I am still a bit concerned about the correctness and efficiency in more general cases, so I will keep my rating.

---

> > > ### Author Response · Authors · 2026-04-06
> > >
> > > **Thank you for the follow-up**. We understand your remaining concern as whether our conclusions should be interpreted as a robust structural phenomenon or only as an effect observed in a narrow setting. We agree this distinction matters.
> > >
> > > Our intended claim is narrower than a universal efficiency claim. The **main contribution** of the paper is the mechanistic explanation of why the first token becomes an attention sink: causal value aggregation creates a positional variance asymmetry, which is then amplified into dimension disparity and sink formation. We believe this structural claim is now supported beyond a single pretrained checkpoint: it appears already at random initialization, persists through training dynamics, and the same qualitative mechanism is reproduced beyond the original setting, including Llama-3-8B/GQA and a GPT-style decoder.
> > >
> > > For the practical intervention, we agree that broader evidence is needed before making strong general claims. Accordingly, we view **HeadNorm as a proof-of-concept intervention**: it demonstrates that directly controlling the identified variance discrepancy improves optimization in the tested settings, rather than claiming universal superiority across all decoder architectures and scales.
> > >
> > > To address robustness more directly, we have added three **independent runs on the exact Section 5 setup** (seeds 0, 1000, 2000). Across seeds, HeadNorm consistently improves:
> > >
> > > **Train loss:** $2.7483 \pm 0.0118 \rightarrow 2.7073 \pm 0.0095$
> > >
> > > **Validation loss:** $2.7812 \pm 0.0109 \rightarrow 2.7421 \pm 0.0066$
> > >
> > > **Effective rank** (layer-wise mean): $343.71 \pm 15.63 \rightarrow 445.96 \pm 5.37$
> > >
> > > **Dimension disparity** (layer-wise mean): $82.67 \pm 8.09 \rightarrow 33.74 \pm 2.73$
> > >
> > > The gaps are materially larger than the across-seed standard deviations, which directly reduces the concern that the reported efficiency gain could be a favorable-run artifact.
> > >
> > > We also want to be explicit about scope: we are **not** claiming that HeadNorm is strictly expressivity-preserving in the formal sense. It removes one scalar norm degree of freedom per head/token. Our claim is instead empirical and targeted: the removed component is precisely the unstable magnitude factor induced by the variance discrepancy, and in our experiments this does **not** create a practical capacity bottleneck; rather, we observe higher effective rank together with better training and validation convergence.
> > >
> > > In the revision, we will make this scope explicit and tone down any wording that could be read as a universal efficiency claim. We hope that, under this scoped interpretation, the reviewer can view the paper’s central contribution—the structural explanation of attention sink formation—as well supported, while treating larger-scale deployment questions as an important downstream direction rather than a blocker for the main finding.

---

### Decision · Program_Chairs · 2026-04-30

**Decision:**

Accept (regular)

**Comment:**

The paper studies the structural origin of attention sinks. Causal value aggregation provides positional variance asymmetry. FFN super neurons amplify this into dimension disparity.  The result is QK locking and sink formation. Two controlled interventions reproduce sinks at arbitrary positions, and Head-wise RMSNorm is proposed as a proof-of-concept fix.


Reviewers liked the mechanistic decomposition and the empirical validation. Main concerns:
 * 6Xox and 5D1n raised concerns about positioning relative to prior work,
 * tSoo was concerned about the distribution-vs-instance link, and
 * GZnF questioned generality beyond pretrained checkpoints.

I think the rebuttal handled these well with i) multi-seed runs on the original setup, ii) new experiments on Llama-2 and GPT-style models, and iii) a clearer explanation of how this work differs from prior works that describe what sinks do functionally (this paper instead explains how they structurally form). Please include these in the revision.